# The Optimally Designed Variational Autoencoder Networks for Clustering and Recovery of Incomplete Multimedia Data

**DOI:** 10.3390/s19040809

**Published:** 2019-02-16

**Authors:** Xiulan Yu, Hongyu Li, Zufan Zhang, Chenquan Gan

**Affiliations:** School of Communication and Information Engineering, Chongqing University of Posts and Telecommunications, Chongqing 400065, China; yuxl@cqupt.edu.cn (X.Y.); zhangzf@cqupt.edu.cn (Z.Z.); gancq@cqupt.edu.cn (C.G.)

**Keywords:** feature learning, incomplete multimedia data, fuzzy c-means, variational autoencoder

## Abstract

Clustering analysis of massive data in wireless multimedia sensor networks (WMSN) has become a hot topic. However, most data clustering algorithms have difficulty in obtaining latent nonlinear correlations of data features, resulting in a low clustering accuracy. In addition, it is difficult to extract features from missing or corrupted data, so incomplete data are widely used in practical work. In this paper, the optimally designed variational autoencoder networks is proposed for extracting features of incomplete data and using high-order fuzzy c-means algorithm (HOFCM) to improve cluster performance of incomplete data. Specifically, the feature extraction model is improved by using variational autoencoder to learn the feature of incomplete data. To capture nonlinear correlations in different heterogeneous data patterns, tensor based fuzzy c-means algorithm is used to cluster low-dimensional features. The tensor distance is used as the distance measure to capture the unknown correlations of data as much as possible. Finally, in the case that the clustering results are obtained, the missing data can be restored by using the low-dimensional features. Experiments on real datasets show that the proposed algorithm not only can improve the clustering performance of incomplete data effectively, but also can fill in missing features and get better data reconstruction results.

## 1. Introduction

The rapid development of communication technologies and sensor networks leads to the increase of heterogeneous data. The proliferation of these technologies in communication networks also has facilitated the development of the wireless multimedia sensor network (WMSN) [1]. Currently, multimedia data on WMSNs are successfully used in many applications, such as industrial control [2], target recognition [3] and intelligent traffic monitoring [4].

Nowadays, multimedia sensors produce a great deal of heterogeneous data, which require new models and technologies to process, particularly neural computing [5], to further promote the design and application of WMSNs [6,7]. However, heterogeneous networks and data are often very complex [8,9], which consist of structured data and unstructured data such as picture, voice, text, and video. Because heterogeneous data come from many input channels in the real world, these data are typical multimodal data, and there is a nonlinear relationship between them [10]. Different modes usually convey different information [11]. For example, images have many details, such as shadows, rich colors and complex scenes, and use titles to display invisible things like the names of objects in the image [12]. Moreover, different forms have complex relationships. In the real world, most multimedia data suffer from a lot of missing values due to sensor failures, measurement inaccuracy and network data transmission problems [13,14]. These features, especially incompleteness, lead to the widespread use of incomplete data in practical applications [15,16]. Lack of data values will affect the decision process of the application servers for specific tasks [17]. The resulting errors can be important for subsequent steps in data processing. Therefore, the recovery of data missing values is essential for processing big data in WMSNs.

As a fundamental technology of big data analysis, clustering divides objects into different clusters based on different similarity measures, making objects in the same cluster more similar to other objects in different groups [18,19]. They are commonly used to organize, analyze, communicate, and retrieve tasks [20]. Traditional data clustering algorithms focus on complete data processing, such as image clustering [21], audio clustering [22] and text clustering [23]. Recently, heterogeneous data clustering methods have been widely concerned by researchers [24,25,26]. In addition, many algorithms have been proposed—for example, Meng et al. optimized the unified objective function by an iterative process, and a spectral clustering algorithm is developed for clustering heterogeneous data based on graph theory [27]. Li et al. [28] proposed a high-order fuzzy c-means algorithm to extend the conventional fuzzy c-means algorithm from vector space to tensor space. A high-order possibilistic c-means algorithm based on tensor decompositions was proposed for data clustering in Internet of Things (IoT) systems [29]. These algorithms are effective to improve clustering performance for heterogeneous data. However, they can only obtain clustering results and lack further analysis of incomplete data low-dimensional features. Therefore, their performance is limited with the heterogeneous data in the WMSNs’ big data environment. More importantly, other existing feature clustering algorithms do not consider data reconstruction and missing data. WMSN systems require different modern data analysis methods, and deep learning (DL) has been actively applied in many applications due to its strong data feature extraction ability [30]. Deep embedded clustering (DEC) learns to map from data space to low-dimensional feature space, where it optimizes the clustering objectives [31]. Ref. [32] shows the feature representation ability of variational autoencoder (VAE). VAE learns the multi-faceted structure of data and achieves high clustering performance [33]. In addition, VAE has a strong ability in feature extraction and reconstruction, and it can be a good tool for handling incomplete data.

Aiming at this research object, the variational autoencoder based high-order fuzzy c-means (VAE-HOFCM) algorithm is presented to cluster and reconstruction incomplete data in WMSNs in this paper. It can effectively cluster complete data and incomplete data and get better reconstruction results. VAE-HOFCM is mainly composed of three steps: feature learning and extraction, high-order clustering, and data reconstruction. First, the feature learning network is improved by using a variational autoencoder to learn the feature of incomplete data. To capture nonlinear correlations of different heterogeneous data, tensors are applied to form a feature representation of heterogeneous data. Then, the tensor distance is used as the distance measure to capture the unknown distribution of data as much as possible in the clustering process. The results of feature clustering and VAE output both affect the final clustering results. Finally, in the case of clustering results, the missing data can be restored by the low-dimensional features.

The rest of the paper is organized as follows: Section 2 presents related work to this paper. The proposed algorithm is illustrated in Section 3, and experimental results and analysis are described in Section 4. Finally, the whole paper is concluded in the last section.

## 2. Preliminaries

This section describes the variational autoencoder (VAE) and the fuzzy c-means (FCM), which will be useful in the sequel.

### 2.1. Variational Autoencoder

The variational autoencoder, which is a new method for nonlinear dimensionality reduction, is a great case of combining probability plots with deep learning [34,35]. Consider a dataset $X=\left\{x_{1},x_{2},\ldots ,x_{N}\right\}$ which consists of *N* independent and identically distributed samples of continuous or discrete variables *x*. To generate target data *x* from hidden variable *z*, two blocks are used: encoder block and decoder block. Suppose that *z* is generated by some prior normal distribution $p_{\theta }=N\left(\mu ,\sigma^{2}\right)$.

The true posterior density $p_{\theta }\left(z\left|x\right.\right)$ is intractable. Approximate recognition model $q_{\phi }\left(z\left|x\right.\right)$ as a probabilistic encoder. Similarly, refer to $p_{\theta }\left(x\left|z\right.\right)$ as a probability decoder because, given the code *z*, it produces a distribution over the possible corresponding value *x*. The parameters θ and ϕ are used to represent the structure and weight of the neural network used. These parameters are adjusted as part of the VAE training process and are considered constant later. Minimize the true posterior approximation of the KL divergence (Kullback–Leibler Divergence). When the divergence of KL is zero, $p_{\theta }\left(z\left|x\right.\right)=q_{\phi }\left(z\left|x\right.\right)$. Then, the true posterior distribution can be obtained. The KL divergence of approximation from the true posterior $D_{KL}\left(q_{\phi }\left(z\left|x\right.\right)∥p_{\theta }\left(z\left|x\right.\right)\right)$ can be formulated as:(1)$$\begin{array}{c} \left(q_{\phi }\left(z\left|x\right.\right)∥p_{\theta }\left(z\left|x\right.\right)\right)=\int_{-\infty }^{\infty }q_{\phi }\left(z\left|x\right.\right)\log\frac{q_{\phi }\left(z\left|x\right.\right)}{p_{\theta }\left(z\left|x\right.\right)}dz\\ =\log p_{\theta }\left(x\right)+D_{KL}\left(q_{\phi }\left(z\left|x\right.\right)∥p_{\theta }\left(z\right)\right)-E_{q_{\phi }\left(z\left|x\right.\right)}\left[\log p_{\theta }\left(x\left|z\right.\right)\right]\\ \geq 0, \end{array}$$
which can also be written as:(2)$$\log p_{\theta }\left(x\right)\geq -D_{KL}\left(q_{\phi }\left(z\left|x\right.\right)∥p_{\theta }\left(z\right)\right)+E_{q_{\phi }\left(z\left|x\right.\right)}\left[\log p_{\theta }\left(x\left|z\right.\right)\right].$$

The right half of the inequality is called the variational lower bound on the marginal likelihood of data *x*, and can be written as:(3)$$L\left(\theta ,\phi ;x\right)\geq -D_{KL}\left(q_{\phi }\left(z\left|x\right.\right)∥p_{\theta }\left(z\right)\right)+E_{q_{\phi }\left(z\left|x\right.\right)}\left[\log p_{\theta }\left(x\left|z\right.\right)\right].$$

The second term $E_{q_{\phi }\left(z\left|x\right.\right)}\left[\log p_{\theta }\left(x\left|z\right.\right)\right]$ requires estimation by sampling. A differentiable transformation $g_{\phi }\left(x,\varepsilon \right)$ of an auxiliary noise variable ε is used to reparameterize the approximation $q_{\phi }\left(z\left|x\right.\right)$. Then, form a Monte Carlo estimates of $E_{q_{\phi }\left(z\left|x\right.\right)}\left[\log p_{\theta }\left(x\left|z\right.\right)\right]$:(4)$$E_{q_{\phi }\left(z\left|x\right.\right)}\left[\log p_{\theta }\left(x\left|z\right.\right)\right]=\frac{1}{M}\sum_{m=1}^{M}\log p_{\theta }\left(x\left|z^{m}\right.\right),$$
where $z^{m}=g_{\phi }\left(x,\varepsilon^{m}\right)=\mu +\varepsilon^{m}⊙\sigma$, $\varepsilon^{m}∼N\left(0,I\right)$ and *m* denotes the number of samples.

### 2.2. Fuzzy C-Means Algorithm (FCM)

The fuzzy c-means algorithm (FCM) is a typical soft clustering technique [36,37]. Given a dataset $X=\left\{x_{1},x_{2},\ldots ,x_{N}\right\}$ with *N* objects and *m* observations, fuzzy partition of set *X* into predefined cluster number *c* and the number of clustering centers denoted by $V=\left\{v_{1},v_{2},\ldots ,v_{c}\right\}$. Their membership functions are defined as $u_{ik}=u_{v_{i}}\left(x_{k}\right)$, in which $u_{ik}$ denotes the membership of $x_{k}$ towards the *i* th clustering center and *c* denotes. FCM is defined by a c×m membership matrix $U=\left\{u_{ik}\left|1\leq i\leq c;1\leq k\leq m\right.\right\}$. FCM minimizes the following objective function [38,39] to calculate the membership matrix *U* and the clustering centers *V*:(5)$$J_{m}\left(U,V\right)=\sum_{k=1}^{n}\sum_{i=1}^{c}\left(u_{ik}\right)d^{2}\left(x_{k},v_{i}\right),$$
where every $u_{ik}$ belongs to the interval (0,1), the summary of all the $u_{ik}$ belonging to the same point is one ($\sum_{i=1}^{c}u_{ik}=1$). In addition, none of the fuzzy clusters is empty, neither do any contain all the data $0<\sum_{k=1}^{m}u_{ik}<m,1\leq i\leq c$. Update the membership matrix and clustering centers by minimizing Equation (Equation 5) via the Lagrange multipliers method:(6)$$u_{ik}=\frac{1}{\sum_{j=1}^{c}{\left(d_{ik}/d_{jk}\right)}^{1/\left(m-1\right)}},$$
(7)$$v_{i}=\frac{\sum_{k=1}^{n}u_{ik}^{m}x_{k}}{\sum_{k=1}^{n}u_{ik}^{m}}.$$

In the traditional FCM algorithm, $d_{ik}$ denotes the Euclidean distance between $x_{i}$ and $v_{k}$, and $d_{jk}$ denotes the Euclidean distance between $x_{j}$ and $v_{k}$.

## 3. Problem Formulation and Proposed Method

Consider a dataset $X=\left\{x_{1},x_{2},\ldots x_{N}\right\}$ with *N* objects. Each object is represented by *m* observations, in the form of $Y=\left\{y_{1},y_{2},\ldots ,y_{m}\right\}$. The purpose of data clustering is to divide datasets into several similar classes based on similarity measure, so that objects in the same cluster have great similarity and are easy to be analyzed. Multimedia data cluster tasks bring many problems and challenges, especially for missing or damaged data. Key challenges are discussed in three areas as below.
Learning the features of incomplete data: feature extraction and analysis are the basic steps of clustering. In general, many feature extraction methods, such as machine learning and deep learning, have been successfully applied to image, text, and audio feature learning. However, the current algorithm focuses on feature learning and extraction of high quality data. In other words, they can not effectively extract the features of lossy data. Therefore, feature learning of incomplete data is the primary problem of heterogeneous data clustering.Clustering in feature space: an important feature of large-scale multimedia data is its diversity, which means that large-scale data sources are diverse, including structured, unstructured data and semi-structured data from a large number of sources. In particular, a large number of objects in large data sets are multi-model. For example, web pages usually contain both images and text. Each mode of multimodal object has its own characteristics, which leads to the complexity of data. Therefore, the feature representation of multimedia data is significant in cluster tasks.Filling missing values to reconstruct data: in wireless multimedia sensor networks, reliable data transmission is critical to provide the ideal quality of network-based services. However, multimedia data transmission may not be successful due to different reasons such as sensory errors, connection errors, or external attacks. These problems can result in incomplete data and degrade the performance of WMSNS applications. After feature extraction and cluster analysis, it is very important to recover missing data from the sensor network.

### 3.1. Description of the Proposed Method

The variational autoencoder based high-order fuzzy c-means (VAE-HOFCM) algorithm is divided into three stages: unsupervised feature learning, high-order feature clustering, and data reconstruction. Architecture of the proposed method is shown in Figure 1.

To learn the features of incomplete multimedia data, the original data set is divided into two different subsets $X_{c}$ and $X_{inc}$. Samples in subset $X_{c}$ have no missing values while each sample contains some missing values in subset $X_{inc}$.

### 3.2. Feature Learning Network Architecture

For trained variational autoencoder, $q_{\phi }\left(z\left|x\right.\right)$ will be very close to $p_{\theta }\left(z\left|x\right.\right)$, so the encode network can reduce the dimensionality of the real dataset $X=\left\{x_{1},x_{2},\ldots ,x_{N}\right\}$ and obtain low-dimensional distribution. In this case, the potential variables may get better results than the traditional dimensionality reduction methods. When the improved VAE model is obtained, the encode network is used to learn the potential feature vectors of missing value sample $z=Encoder\left(x\right)∼q_{\phi }\left(z\left|x\right.\right)$. The decode network is then used to decode the vector *z* to generate the original sample $\bar{x}=Decoder\left(z\right)∼p_{\theta }\left(x\left|z\right.\right)$.

According to the original VAE and to build a better generation model, convolution kernels are added to the encoder. There is a variational constraint on the latent variable *z*, that is, *z* obeys the Gauss distribution. Here, each $x_{i}\left(1\leq i\leq N\right)$ is fitted with an exclusive normal distribution. Sample *z* is then extracted from the exclusive distribution, since $z_{i}$ is sampled from the exclusive $x_{i}$ distribution, the original sample $x_{i}$ can be generated through a decoder network. The improved VAE model is shown in Figure 2.

In general, assume that $q_{\phi }\left(z\right)$ is the standard normal distribution, $q_{\phi }\left(z\left|x\right.\right)$, $p_{\theta }\left(x\left|z\right.\right)$ are the conditional normal distribution, and then plug in the calculation to get the normal loss of VAE, where *z* is a continuous variable representing the coding vector, and *y* is a discrete variable that represents a category. If *z* is directly replaced in the formula with $\left(z,y\right)$, the loss of the clustered VAE is obtained:(8)$$\;D_{KL}\left(q_{\phi }\left(z,y\left|x\right.\right)∥p_{\theta }\left(z,y\left|x\right.\right)\right)=\int_{-\infty }^{\infty }q_{\phi }\left(z,y\left|x\right.\right)\log\frac{q_{\phi }\left(z,y\left|x\right.\right)}{p_{\theta }\left(z,y\left|x\right.\right)}dz.$$

Set the scheme as: $q_{\phi }\left(z,y\left|x\right.\right)=q_{\phi }\left(y\left|z\right.\right)q_{\phi }\left(z\left|x\right.\right)$, $p_{\theta }\left(x\left|z,y\right.\right)=p_{\theta }\left(x\left|z\right.\right)$, $p_{\theta }\left(z,y\right)=p_{\theta }\left(z\left|y\right.\right)p_{\theta }\left(y\right)$. Substituting them into Equation (Equation 8) and it can be simplified as follows:(9)$$\;E_{q_{\phi }\left(x\right)}\left[-\log p_{\theta }\left(x\left|z\right.\right)+\sum_{y}q_{\phi }\left(y\left|z\right.\right)D_{KL}\left(q_{\phi }\left(z\left|x\right.\right)∥p_{\theta }\left(z\left|y\right.\right)\right)+D_{KL}\left(q_{\phi }\left(y\left|z\right.\right)∥p_{\theta }\left(y\right)\right)\right],$$
where the first term $-\log p_{\theta }\left(x\left|z\right.\right)$ wants the reconstruction error to be as small as possible, that is, *z* keeps as much information as possible. $\sum_{y}q_{\phi }\left(y\left|z\right.\right)D_{KL}\left(q_{\phi }\left(z\left|x\right.\right)∥p_{\theta }\left(z\left|y\right.\right)\right)$ plays the role of clustering. In addition, $D_{KL}\left(q_{\phi }\left(y\left|z\right.\right)∥p_{\theta }\left(y\right)\right)$ makes the distribution of each class as balanced as possible; there will not be two nearly overlapping situations. The above equation describes the coding and generation process:Sampling to *x* from the original data, coding feature *z* can then be obtained by $q_{\phi }\left(z\left|x\right.\right)$. Then, the coding feature is classified by classifier $q_{\phi }\left(y\left|z\right.\right)$ to obtain the classification.Select a category *y* from distribution $p_{\theta }\left(y\right)$, select a random hidden variable *z* from distribution $p_{\theta }\left(z\left|y\right.\right)$, and then decode the original sample through generator $p_{\theta }\left(x\left|z\right.\right)$.

The VAE is outlined in Algorithm 1.

**Algorithm 1** Variational Autoencoder Optimization.**Input:** Training set $X={\left\{x_{t}\right\}}_{t=1}^{N}$, corresponding labels $Y={\left\{y_{t}\right\}}_{t=1}^{N}$, loss weight $\lambda_{1},\lambda_{2},\lambda_{3}$.
**Output:** VAE parameters θ,ϕ.
1:**Initialization:** random initialized $\theta_{0},\phi_{0}$.2:**Repeat:**    Sample $x_{t}$ in the minibatch.3:   $\mu_{z_{t}}=Encoder\left(x_{t}\right)∼q_{\phi }\left(z\left|x\right.\right)$4:   Sample: $z_{t}\leftarrow \mu_{z_{t}}+\varepsilon ⊙\sigma_{z}$, $\varepsilon ∼N\left(0,I\right)$5:   $\mu_{x_{t}}=Decoder\left(z_{t}\right)∼p_{\theta }\left(x\left|z\right.\right)$6:   Compute reconstruction loss: $L_{rec}=-\log p_{\theta }\left(x_{t}\left|z_{t}\right.\right)$.7:   Compute regularization loss: $L_{reg}=D_{KL}\left(q_{\phi }\left(y_{t}\left|z_{t}\right.\right)∥p_{\theta }\left(y_{t}\right)\right)$.8:   Compute clustering loss: $L_{cls}=\sum_{y}q_{\phi }\left(y_{t}\left|z_{t}\right.\right)D_{KL}\left(q_{\phi }\left(z_{t}\left|x_{t}\right.\right)∥p_{\theta }\left(z_{t}\left|y_{t}\right.\right)\right)$.9:   Fuse the three loss: $L\left(\theta ,\phi \right)=\lambda_{1}L_{rec}\left(\theta ,\phi \right)+\lambda_{2}L_{reg}\left(\theta ,\phi \right)+\lambda_{3}L_{cls}\left(\theta ,\phi \right)$.10:   Back-propagate the gradients.11:**Until** maximum iteration reached.


### 3.3. Variational Autoencoder Based High-Order Fuzzy C-Means Algorithm

Variational autoencoder gets the low-dimensional features and initial clustering results of data by feature learning. Then, the final clustering results will be optimized by the FCM algorithm clustering results. Traditional FCM work in vector space. It is better to use higher-order tensor to represent the feature of data because the tensor distance can capture the correlation in the high-order tensor space and measures the similarity between two higher-order complex data samples. Given an *N*-order tensor $X\in R^{I_{1}\times I_{2}\times \ldots \times I_{N}}$, *x* is denoted as the vector form representation of *X*, and the element $X_{i_{1}i_{2}\ldots i_{N}\left(1\leq i_{j}\leq I_{j},1\leq j\leq N\right)}$ in *X* is corresponding to $x_{l}$. That is, the *N* element in *X* is $l=i_{1}+\sum_{j=2}^{N}\prod_{t=1}^{j-1}I_{t}$. Then, the tensor distance between two *N*-order tensors is defined as:(10)$$\;d_{td}=\sqrt{\sum_{l,m=1}^{I_{1}\times I_{2}\times \ldots \times I_{N}}g_{lm}\left(x_{l}-y_{l}\right)\left(x_{m}-y_{m}\right)}=\sqrt{{\left(x-y\right)}^{T}G\left(x-y\right)},$$
where $g_{lm}$ is the metric coefficient and used to capture the correlations between different coordinates in the tensor space, which can be calculated by:(11)$$\;g_{lm}=\frac{1}{2\pi \delta^{2}}\exp\left\{-\frac{{∥p_{l}-p_{m}∥}_{2}^{2}}{2\delta^{2}}\right\},$$
where ${∥p_{l}-p_{m}∥}_{2}$ is defined as:(12)$$\;{∥p_{l}-p_{m}∥}_{2}=\sqrt{{\left(i_{1}-{i_{1}}^{\prime }\right)}^{2}+\cdots +{\left(i_{N}-{i_{N}}^{\prime }\right)}^{2}}.$$

Minimizing the objective function of high-order fuzzy c-means algorithm:(13)$$\;J_{m}\left(U,V\right)=\sum_{k=1}^{n}\sum_{i=1}^{c}\left(u_{ik}\right)d_{td}^{2}.$$

To update the membership value $u_{ik}$, we differentiate with respect to $u_{ik}$, as follows:(14)$$\;\begin{array}{c} \frac{\partial J_{m}\left(U,V\right)}{\partial u_{ij}}=\frac{\partial \left({\left(u_{ik}\right)}^{m}d_{td}^{2}\left(x_{k},v_{i}\right)\right)}{\partial u_{ij}}\\ =m\cdot {\left(u_{ij}\right)}^{m-1}d_{td}^{2}\left(x_{j},v_{i}\right). \end{array}$$

Setting Equation (Equation 14) to 0, $u_{ik}$ is calculated:(15)$$\;u_{ik}=\frac{1}{\sum_{j=1}^{c}{\left(\frac{d_{\left(td\right)ik}}{d_{\left(td\right)jk}}\right)}^{1/\left(m-1\right)}}.$$

Then, the equation for updating $v_{i}$ is obtained:(16)$$\;v_{i}=\frac{\sum_{j=1}^{n}u_{ij}^{m}x_{j}}{\sum_{j=1}^{n}u_{ij}^{m}}.$$

For each iteration, this operation requires $O\left(c\times n\right)$, so the total computational complexity of *k* iterations is $O\left(kc\times n\right)$. From the above, the VAE-HOFCM algorithm can be described as Algorithm 2:

**Algorithm 2** The VAE-HOFCM algorithm.**Input:**X=x1,x2,…,xn**Output:**U=uij and V=vi.
1:Initialize X=x1,x2,…,xn randomly.2:Perform Algorithm 1 to calculate low dimensional representation of dataset *X*: x=Encoderxn3:**for**iteration=1,2,…,maxiter4:   **for:**
i=1,2,…,c5:     vi=∑j=1nuijmxj∑j=1nuijm6:   **for:**
i=1,2,…,c7:     **for:**
j=1,2,…,c8:      $u_{ij}=\frac{1}{\sum_{j=1}^{c}{\left(\frac{d_{\left(td\right)ik}}{d_{\left(td\right)jk}}\right)}^{1/\left(m-1\right)}}$9:$\left(x,y\right)=Decoder\left(z_{t}\right)$.10:Obtain the modified clustering results using the $u_{ij}$.


By comparing the steps of the HOFCM algorithm, VAE-HOFCM can restore incomplete data simultaneously in the clustering process. Equally, the VAE-HOFCM algorithm has a total time complexity of $O\left(kc\times n\right)$. However, before that, it needs to train the variational autoencoder network.

## 4. Experiments

This section evaluates the performance of the proposed VAE-HOFCM algorithm on three representative datasets. To show the effectiveness of VAE-HOFCM, the unsupervised clustering accuracy (ACC) and adjusted rand index (ARI) for verification are adopted. ACC is calculated by:(17)$$\;ACC=\max_{m}\frac{\sum_{i=1}^{n}1\left\{l_{i}=m\left(c_{i}\right)\right\}}{n},$$
where $l_{i}$ and $c_{i}$ indicate the ground-truth label and the cluster assignment produced by the algorithm, respectively. *m* ranges overall possible one-to-one mappings between clusters and labels. ARI is used to measure the agreement between two possibilistic partitions of a set of objects, where *U* denotes the true labels of the objects in datasets, and $U^{\prime }$ denotes a cluster generated by a specific algorithm. A higher value of $ARI\left(U,U^{\prime }\right)$ represents that the algorithm has more accurate clustering results.

To study the performance and generality of different algorithms, experiments are performed on three datasets:MNIST: The MNIST dataset consists of 70,000 hand-written digits of 28-by-28 pixel size. The digits are centered and and the size is standardized.STL-10: A dataset consists of 96-by-96 color images. It contains 13,000 labeled images and 100,000 unlabeled images.NUS-WIDE: The NUS-WIDE dataset consists of 269,648 images and can be downloaded from Flickr.com, a famous photo-sharing website.

### 4.1. Experimental Results on Complete Datasets

This section evaluates the performance of variational autoencoder based high-order fuzzy c-means algorithm (VAE-HOFCM) in clustering compared to other algorithms. The input dimensions of these three datasets are 784, 3072 and 500, respectively. The dimension of VAE hidden layer is set as 25, and the number of training iterations of the training set as 50. After obtaining the low-dimensional features, start clustering, and the membership factor is set as 2.5. Then, the required clustering center is calculated and the final normalized membership matrix *U* is returned to obtain the clustering result.

The clustering results are shown in Table 1 and Table 2. Table 1 displays the optimal performance of unsupervised clustering accuracy of each algorithm. For MNIST data clustering class, the proposed VAE-HOFCM algorithm has achieved the highest accuracy of 85.54%. Compared with VAE clustering, the VAE-HOFCM encoder training time and cluster running time sum is slightly more than the former, but the clustering accuracy is improved. Then, the clustering performance and running time of VAE-HOFCM algorithm are generally better than traditional clustering algorithms, such as k-means and fuzzy c-means. Since the dimension of STL-10 dataset is higher and the information content is larger, the operation time of extracting features and clustering is relatively long. However, the proposed algorithm still gets the best running results. Visual features and text features are extracted from the NUS-WIDE dataset, and then these features are connected to form feature vectors. Finally, the feature vectors are clustered. The clustering results show the performance of the proposed algorithm.

Table 2 shows the clustering results in terms of $ARI\left(U,U^{\prime }\right)$, VAE-HOFCM produces high value than other algorithms in most cases. K-means usually has the worst performance and the longest running time, whereas VAE and DEC achieve the better result than HOPCM. ARI is not used as an indicator in the STL-10 dataset because the value may be negative in the case of clustering accuracy.

There are two reasons for the results of these results in terms of ACC and ARI. On the one hand, HOFCM integrates the learning characteristics of different modes, uses the cross product to model the nonlinear correlation under various modes, and uses the tensor distance as a measure to capture the high-dimensional distribution of multimedia data. On the other hand, VAE successfully learns low-dimensional features and achieves the best performance in feature dimension reduction and clustering accuracy.

VAE has good data clustering and data generation performance. Feature extraction is carried out by the VAE to reduce the dimension to two dimensions. These categories have clear boundaries as shown in Figure 3, indicating that the VAE has effectively extracted low-dimensional features. This proves that the VAE has strong data feature expression ability.

To obtain better performance in the three constraints of data feature dimension, clustering performance and reconstruction quality, the quality of data reconstruction in different dimensions is compared. Figure 4 shows the reproduction performance of learning generation models for different dimensions. When the latent space is set at 25, this method can obtain a good reconstruction quality.

Figure 5 shows the generated images of two clustering results categories 1 and 6 of MNIST.

### 4.2. Experimental Results on Incomplete Data Sets

To estimate the robustness of the proposed algorithm, each dataset is divided into complete datasets and incomplete datasets. Now, incomplete datasets are used for simulation analysis. Since clustering performance depends on the number of missing values, six miss rates are set, which are 5%, 10%, 15%, 20%, 25% and 30%, respectively.

Figure 6 shows the clustering results accuracy of ACC with the increase of the missing ratio on the MNIST dataset and NUS-WIDE dataset. Figure 7 shows the average values of ARI with the increase of the missing ratio on the MNIST dataset and NUS-WIDE dataset. The results show that the increase of missing rate will lead to the decrease of clustering accuracy. However, the proposed algorithm still has a high accuracy because VAE successfully extracts incomplete data features and reduces the difference with the incomplete data features.

According to Figure 6 and Figure 7, with the increase of missing rate, the average value of ACC and ARI would decrease, which indicates that the missing rate destroys the original data content, leading to the decrease of clustering accuracy. The average ACC and ARI values based on the VAE-HOFCM algorithm are significantly higher than those of the other three methods at the six missing rates. Therefore, VAE-HOFCM clustering has the best performance, indicating that VAE-HOFCM is also effective for clustering incomplete data.

Then, data with different missing rates are reconstructed, as shown in Figure 8. Inputs are incomplete data with different missing rates, and the output are recovered data using VAE. The reconstruction results show that the proposed algorithm not only improves the clustering accuracy, but also ensures that the data can be reconstructed with high quality.

The variational auto-coder also has the function of de-noising. As shown in Figure 9, noise is added into the input data to enable VAE to effectively de-noise and restore the original input image.

## 5. Conclusions

In this paper, a VAE-HOFCM algorithm, which can improve the performance of multimedia data clustering, has been proposed. Unlike many existing technologies, the VAE-HOFCM algorithm learns the data features by designing an improved VAE network, and uses a tensor based FCM algorithm to cluster the data features in the feature space. In addition, VAE-HOFCM captures as many features of high quality multimedia data and incomplete multimedia data as possible. In experiments, the performance of the proposed scheme has been evaluated on three heterogeneous datasets, MNIST, STL-10 and NUS-WIDE. Compared with traditional clustering algorithms, the results show that VAE can achieve a high compression rate of data samples, save memory space significantly without reducing clustering accuracy, and enable low-end devices in wireless multimedia sensor networks to achieve clustering of large data. In addition, VAE can effectively fill the missing data and generate the specified data at the terminal, so that the incomplete data can be better utilized and analyzed. Although VAE needs to be trained well, the sum time of training and clustering is still less than most clustering algorithms. Therefore, when performing clustering tasks on low-end equipment with limited computing power and memory space, trained VAE-HOFCM can be adopted.

## Figures and Tables

**Figure 1 sensors-19-00809-f001:**
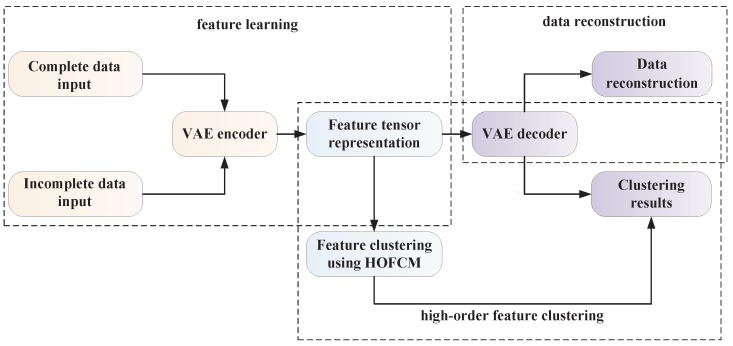
Architecture of the proposed method.

**Figure 2 sensors-19-00809-f002:**
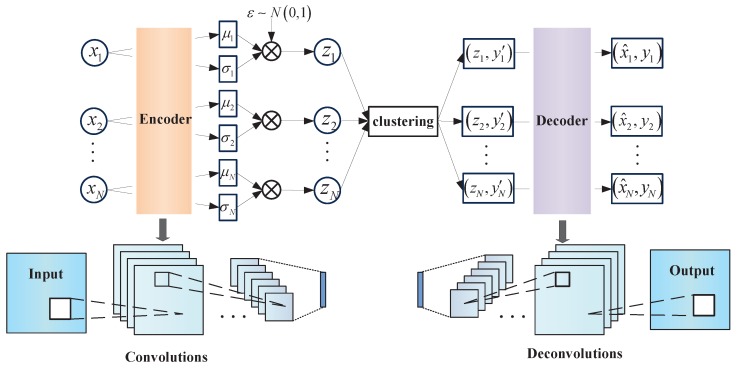
The improved VAE model.

**Figure 3 sensors-19-00809-f003:**
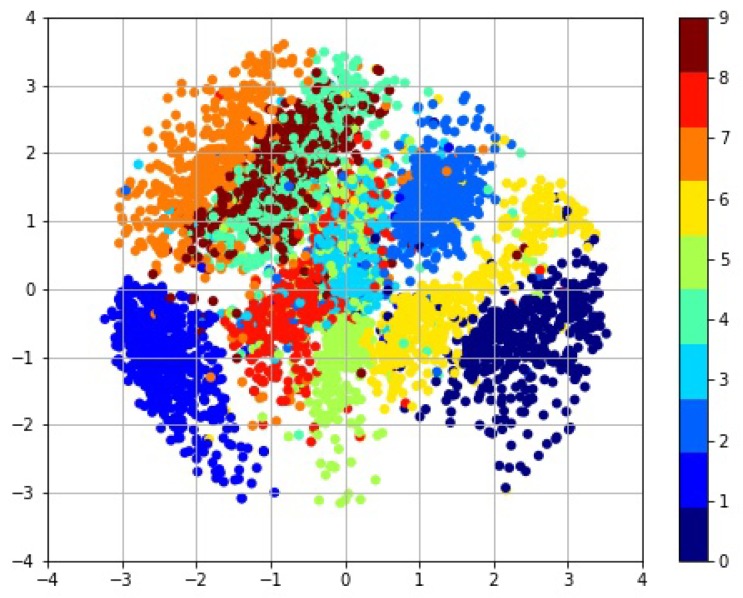
Visual analysis of MNIST datasets.

**Figure 4 sensors-19-00809-f004:**

Reconstruction quality for different dimensionalities.

**Figure 5 sensors-19-00809-f005:**
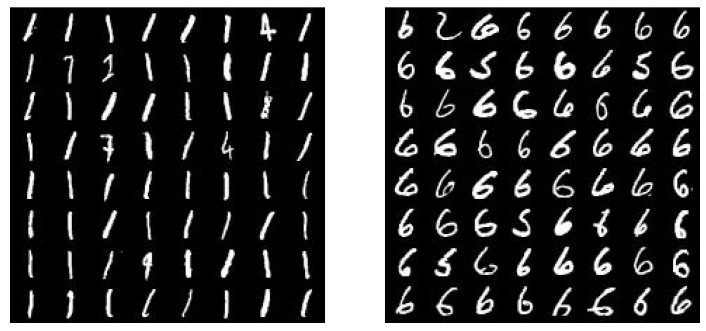
Cluster category sampling.

**Figure 6 sensors-19-00809-f006:**
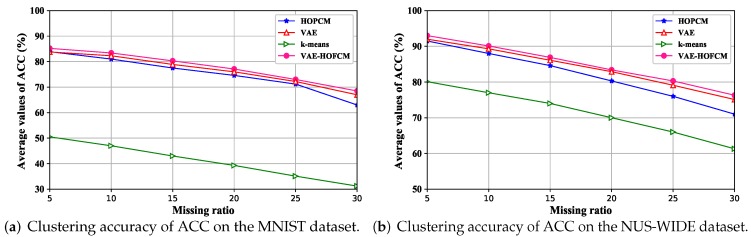
Clustering accuracy of ACC.

**Figure 7 sensors-19-00809-f007:**
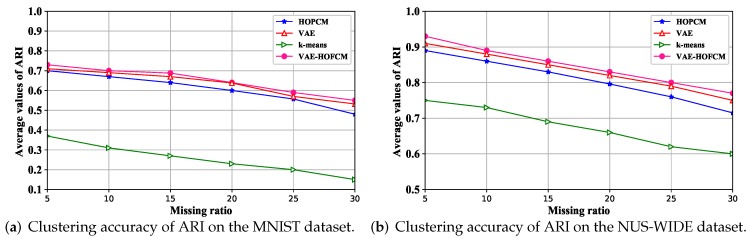
Clustering accuracy of ARI.

**Figure 8 sensors-19-00809-f008:**
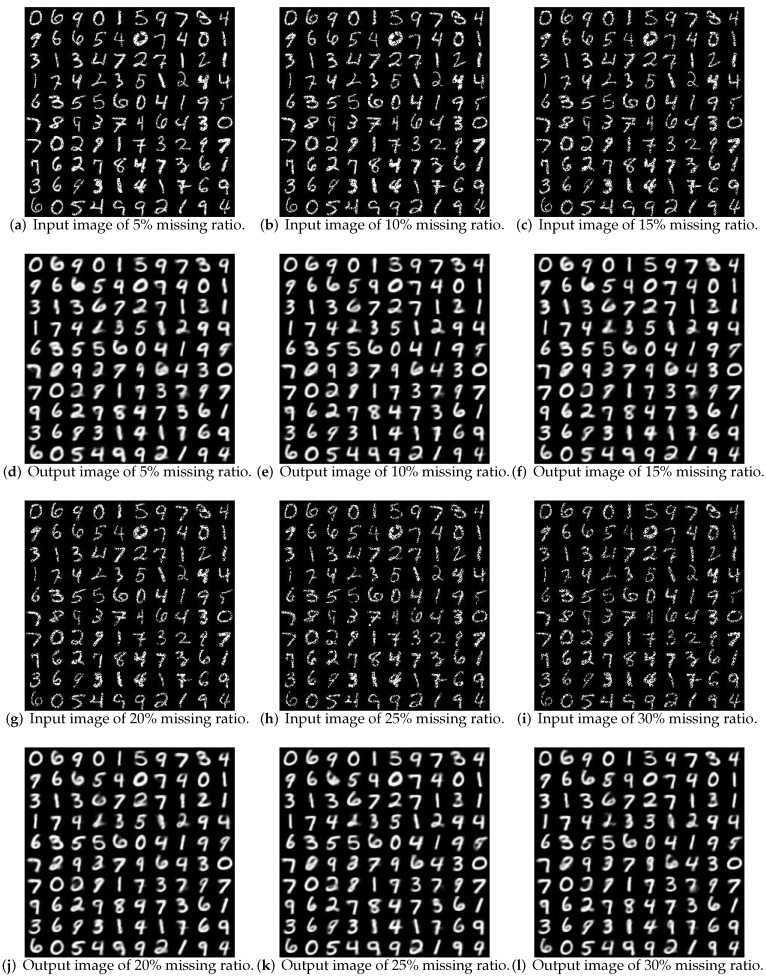
Reconstruction quality for different dimensionalities.

**Figure 9 sensors-19-00809-f009:**

Reconstruction quality for noise data.

**Table 1 sensors-19-00809-t001:** Clustering accuracy of ACC.

Algorithm/Dataset	MNIST	STL-10	NUS-WIDE
k-means	53.49%	28.40%	81.51%
HOPCM	80.34%	33.12%	92.75%
VAE	84.20%	35.48%	93.32%
DEC	84.31%	35.90%	93.75%
VAE-HOFCM	85.54%	36.44%	95.14%

**Table 2 sensors-19-00809-t002:** Clustering accuracy of ARI.

Algorithm/Dataset	MNIST	STL-10	NUS-WIDE
k-means	0.41	-	0.74
HOPCM	0.69	-	0.89
VAE	0.75	-	0.90
DEC	0.76	-	0.90
VAE-HOFCM	0.78	-	0.92

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
