# Peer review of "The Optimally Designed Variational Autoencoder Networks for Clustering and Recovery of Incomplete Multimedia Data"

_sensors, 2019, doi:10.3390/s19040809_

Reviewer 1 Report

Abstract is clear.

Introduction is a little extensive, but explain well the studied problem and the state-of-the-art clustering algorithms.

Section 2.1 “Variational Autoencoder” some reference must be provided for a better understanding about that. Any reference is provided on this section.

Section 2.1: it is difficult understand all variables on equations 1 to 4 without any reference about that. All variables must be defined as for example theta, phi, M.

Section 2.2, the same problem about the references than Section 2.1. There is any reference to understand how equations 5 to 7 are obtained.

On section 2.1 the dataset is defined in one way and on section 2.2 in other way, should be the same. The same suggestion for all variables presented on the paper.

Section 2.2 m is defined as attributes, it seems that is not well defined.

The steps of FCM are not clear. A flowchart/pseudo-code must be present to be more clear the FCM algorithm.

Section 2.2, how are defined the membership functions (MFS)? Equation of sigma ( MFs’ with) is not defined.

Some suggested reference to improve section 2.2:

- Celikyilmaz, A., & Trksen, I. B. (2009). Modeling Uncertainty with Fuzzy Logic: With Recent Theory and Applications. (1st ed.). Springer Publishing Company, Incorporated.

-  Dovžan, D., & Škrjanc, I. (2011). Recursive fuzzy c-means clustering for recursive fuzzy identification of time-varying processes. ISA Transactions, 50, 159–169.

- Jérôme Mendes, Rui Araújo, Francisco Souza. Adaptive fuzzy identification and predictive control for industrial processes. Expert Systems with Applications, v. 40, n. 17,p. 6964-6975, December 2013.

- Saeid Rastegar, Rui Araújo, and Jérôme Mendes. Online Identification of Takagi–Sugeno Fuzzy Models Based on Self-Adaptive Hierarchical Particle Swarm Optimization Algorithm. Applied Mathematical Modelling, v.45, p. 606-620, May 2017.

Section 3, dataset is defined on a different way than sections 2.1 and 2.2. The same for sections 3.2

Section 3, on Figure 1 should be more clear the 3 stages of the proposed method (unsupervised feature learning, high-order feature clustering, and data reconstruction)

Section 3.2, all the steps of Algorithm 1 (steps 3 to 10) are not easy to understand their implementation, since they not contain the respective equations or reference to how to be implement.

Author Response

Response to Reviewer 1 Comments

 The authors are grateful to the anonymous reviewers for their valuable comments and suggestions, which are highly insightful and enable us to greatly improve the quality of our manuscript. The comments are well taken and the manuscript has been revised accordingly. The revisions are highlighted in yellow in the paper. Our point-by-point responses to the comments are given as follows:

Point 1: Section 2.1 “Variational Autoencoder” some reference must be provided for a better understanding about that. Any reference is provided on this section. Section 2.1: it is difficult understand all variables on equations 1 to 4 without any reference about that. All variables must be defined as for example theta, phi, M.

 Response 1: We cited references and explained variables.

 Point 2: Section 2.2, the same problem about the references than Section 2.1. There is any reference to understand how equations 5 to 7 are obtained..

 Response 2: We cite the references you suggested and explained the formulas and parameters in detail.

Point 3: On section 2.1 the dataset is defined in one way and on section 2.2 in other way, should be the same. The same suggestion for all variables presented on the paper.

 Response 3: We uniformly define dataset.

 Point 4: Section 2.2 m is defined as attributes, it seems that is not well defined.

 Response 4: m is redefined as observations.

 Point 5: The steps of FCM are not clear. A flowchart/pseudo-code must be present to be more clear the FCM algorithm.

 Response 5: By referring to the references you have given, we have made some modifications and demonstrated some of the core formulas and architectures. Besides, the references have been updated.

 Point 6: Section 2.2, how are defined the membership functions (MFS)? Equation of sigma ( MFs’ with) is not defined.

 Response 6: We supplemented the definition.

 Point 7: Section 3, dataset is defined on a different way than sections 2.1 and 2.2. The same for sections 3.2

 Response 7: We uniformly define dataset.

 Point 8: Section 3, on Figure 1 should be more clear the 3 stages of the proposed method (unsupervised feature learning, high-order feature clustering, and data reconstruction)

 Response 8: We further revised Figure 1 to make it clearer..

 Point 9: Section 3.2, all the steps of Algorithm 1 (steps 3 to 10) are not easy to understand their implementation, since they not contain the respective equations or reference to how to be implement.

 Response 9: The formulas in the algorithm table have been modified and the contents have been added.

Reviewer 2 Report

The authors deal with the problem of improving the cluster performance of incomplete data and propose the optimally designed variational autoencoder networks for extracting features of incomplete data and using high-order fuzzy c-means algorithm in order to improve cluster performance of incomplete data. Moreover, the authors present a set of experiments validating their claims.

The article is interesting, written by experts in the field, while the whole presented algorithmic concept seems of interest to prospective researchers and hence, I vote for acceptance. 

Author Response

The authors are grateful to the anonymous reviewers for their valuable comments and suggestions。